# Urinary Mass Spectrometry Profiles in Age-Related Macular Degeneration

**DOI:** 10.3390/jcm11040940

**Published:** 2022-02-11

**Authors:** Ines Lains, Kevin M. Mendez, João Q. Gil, John B. Miller, Rachel S. Kelly, Patrícia Barreto, Ivana K. Kim, Demetrios G. Vavvas, Joaquim Neto Murta, Liming Liang, Rufino Silva, Joan W. Miller, Jessica Lasky-Su, Deeba Husain

**Affiliations:** 1Massachusetts Eye and Ear, Department of Ophthalmology, Harvard Medical School, Boston, MA 02114, USA; ines_lains@meei.harvard.edu (I.L.); john_miller@meei.harvard.edu (J.B.M.); ivana_kim@meei.harvard.edu (I.K.K.); demetrios_vavvas@meei.harvard.edu (D.G.V.); joan_miller@meei.harvard.edu (J.W.M.); 2Systems Genetics and Genomics Unit, Channing Division of Network Medicine, Brigham and Women’s Hospital and Harvard Medical School, Boston, MA 02115, USA; kevin_mendez@meei.harvard.edu (K.M.M.); hprke@channing.harvard.edu (R.S.K.); jessica.su@channing.harvard.edu (J.L.-S.); 3Faculty of Medicine, University of Coimbra, 3000-548 Coimbra, Portugal; joaomqgil@gmail.com (J.Q.G.); jmurta@netcabo.pt (J.N.M.); rufino.silva@oftalmologia.co.pt (R.S.); 4Association for Innovation and Biomedical Research on Light and Image, 3000-548 Coimbra, Portugal; pbarreto@aibili.pt; 5Ophthalmology Department, Centro Hospitalar e Universitário de Coimbra, 3004-561 Coimbra, Portugal; 6Department of Biostatistics, Harvard T.H. Chan School of Public Health, Boston, MA 02115, USA; llinag@hsph.harvard.edu

**Keywords:** age-related macular degeneration, metabolomics, urine

## Abstract

We and others have shown that patients with different severity stages of age-related macular degeneration (AMD) have distinct plasma metabolomic profiles compared to controls. Urine is a biofluid that can be obtained non-invasively and, in other fields, urine metabolomics has been proposed as a feasible alternative to plasma biomarkers. However, no studies have applied urinary mass spectrometry (MS) metabolomics to AMD. This study aimed to assess urinary metabolomic profiles of patients with different stages of AMD and a control group. We included two prospectively designed, multicenter, cross-sectional study cohorts: Boston, US (*n* = 185) and Coimbra, Portugal (*n* = 299). We collected fasting urine samples, which were used for metabolomic profiling (Ultrahigh Performance Liquid chromatography—Mass Spectrometry). Multivariable logistic and ordinal logistic regression models were used for analysis, accounting for gender, age, body mass index and use of AREDS supplementation. Results from both cohorts were then meta-analyzed. No significant differences in urine metabolites were seen when comparing patients with AMD and controls. When disease severity was considered as an outcome, six urinary metabolites differed significantly (*p* < 0.01). In particular, two of the metabolites identified have been previously shown by our group to also differ in the plasma of patients of AMD compared to controls and across severity stages. While there are fewer urinary metabolites associated with AMD than plasma metabolites, this study identified some differences across stages of disease that support previous work performed with plasma, thus highlighting the potential of these metabolites as future biomarkers for AMD.

## 1. Introduction

Age-related macular degeneration (AMD) is one of the leading causes of irreversible adult blindness in the world, [1] and poses a tremendous socioeconomic burden related to an aging population [2]. In its early and intermediate stages, AMD is mostly asymptomatic, and individuals are frequently unaware of their diagnosis except if they have a dilated eye exam. However, some patients progress to the blinding forms of the disease, choroidal neovascularization (neovascular form) or geographic atrophy [3,4]. Even though AMD progression can be predicted based on retinal appearance by a dilated ophthalmic examination, the utility of this prediction is limited. Even wet AMD can go undetected by patients for a long time, especially if they have preserved vision in the fellow eye, with previous reports describing a mean duration of symptom onset to assessment of 2 months [5]. Importantly, when treatment for neovascular AMD is delayed, worse visual functional outcomes have been reported [6].

The identification of biomarkers to assess the risk of AMD would be a major advance, but previous attempts revealed variable results, [7,8] probably due to the complexity of this disease, which involves interactions between genetic and environmental risk factors.

Metabolomics, the qualitative and quantitative analysis of metabolites (<1–1.5 KDa), is an integrative approach that can help address these questions [9]. Metabolites are downstream of the genome and its interaction with environmental exposures. Therefore, the metabolome is thought to closely relate to disease phenotype, especially with multifactorial diseases such as AMD [10]. Metabolomic profiling can be performed using two main analytical tools: nuclear magnetic resonance (NMR) spectroscopy and mass spectrometry (MS) [11]. NMR offers simple sample handling, the possibility of sample reuse, and high reproducibility [12,13]. MS, however, has a much higher sensitivity than NMR, thus enabling the measurement of a broader range of metabolites [10].

As NMR spectroscopy can represent an appropriate technique for an initial untargeted approach, we started our work [14] by using this technique. Using NMR, we observed small differences in both plasma [14] and urine [15] samples between AMD stages. These results motivated us to continue our work with MS, which is becoming the most widely used technology [16]. Our plasma MS results [17,18] revealed that metabolomics enables the identification of specific metabolomic profiles in AMD, which vary with the severity stages. To date, however, neither our group nor others, have used MS to explore potential urinary metabolomic biomarkers of AMD. Even though most research on metabolomics to derive biofluid disease biomarkers has been performed in blood samples, [19,20,21,22] urine has been increasingly used as a biomatrix for metabolomic profiling in other multifactorial diseases [23]. Urine is a very appealing biofluid because it can be collected non-invasively, is produced regularly and in abundant quantities, and gives a time-averaged representation of an individual’s recent homeostatic status [23].

This work aims to assess urinary metabolomic profiles of patients with different stages of AMD and a control group, with the ultimate goal of understanding if urine samples are suitable to identify biomarkers of AMD for this blinding disease.

## 2. Materials and Methods

### 2.1. Study Design

This study was prospectively designed, and it was observational and cross-sectional in nature. We recruited patients from two sites: Boston, US, at the Department of Ophthalmology of Massachusetts Eye and Ear (MEE), Harvard Medical School; Coimbra, Portugal, at the Faculty of Medicine of the University of Coimbra (FMUC), in collaboration with the Association for Innovation and Biomedical Research on Light and Image (AIBILI) and the “Centro Hospitalar e Universitário de Coimbra”. The study was approved by the Institutional Review Boards of FMUC and AIBILI, and by the Portuguese National Data Protection Committee (CNPD), as well as by MEE/Mass General Brigham. The clinical protocol was conducted in accordance with the tenets of the Declaration of Helsinki and with HIPAA (Health Insurance Portability and Accountability Act). All included subjects provided written informed consent.

### 2.2. Study Population

We recruited subjects with a diagnosis of AMD and control subjects with no evidence of AMD (aged ≥ 50 years). Exclusion criteria included: active uveitis or ocular infection, diagnosis of any other vitreoretinal disease, significant media opacities that precluded the observation of the ocular fundus, refractive error equal or greater than 6 diopters of spherical equivalent, past history of retinal surgery, history of any ocular surgery or intra-ocular procedure (such as laser and intra-ocular injections) within the 90 days prior to enrolment, and diagnosis of diabetes mellitus.

### 2.3. Study Protocol

As previously described, [14,15,17] a complete bilateral ophthalmologic examination was performed to all study participants, and all were imaged with 7-field, non-stereoscopic color fundus photographs (CFP), with a Topcon TRC-50DX (Topcon Corporation, Tokyo, Japan) or a Zeiss FF-450Plus (Carl Zeiss Meditec, Dublin, CA, USA) camera. A complete medical history was also obtained, [24] including data on current medications and self-reported body mass index (BMI). REDCap electronic data capture tools was used for data storage. Additionally, urine samples were collected into sterile cups and then stored into sterile cryovials of 1.5 mL (MEE) and 5 mL (FMUC/AIBILI), which were stored at −80 °C. All samples were collected in the morning after confirmed overnight fasting. For those not fasting at the inclusion visit, a new appointment was scheduled for urine collection within a maximum of one month.

### 2.4. AMD Diagnosis and Staging

Images were standardized using software developed by our group prior to grading [25]. Then, two of three independent experienced graders analyzed field 2 CFP, according to the AREDS classification system [26,27]. Cases of disagreement were resolved by a senior author (RS or DH). As reported, [14,15,17] we adopted the most recent AREDS2 definitions [26,27]: controls—presence of drusen maximum size < circle C0 and total area < C1; early AMD—drusen maximum size ≥ C0 but <C1 or presence of AMD characteristic pigment abnormalities in the inner or central subfields; intermediate AMD—presence of drusen maximum size ≥ C1 or of drusen maximum size ≥ C0 if the total area occupied is > I2 for soft indistinct drusen and > O2 for soft distinct drusen; late AMD—presence of GA according to the criteria described above (GA or “dry” late AMD) or evidence of neovascular AMD (choroidal neovascularization, CNV or “wet” AMD).

### 2.5. Mass Spectrometry Analysis

Urine samples from Coimbra, Portugal were shipped to MEE in dry ice. Then, all samples (i.e., from both study locations) were shipped to Metabolon, Inc.^®^ (Morrisville, NC, USA), also in dry ice. In both cases, samples arrived frozen in less than 48 h and were immediately stored at −80 °C until processing. Non-targeted MS analysis was performed by Metabolon, using Ultrahigh Performance Liquid Chromatography–Tandem MS (UPLC-MS/MS), according to protocols that have been previously described [28]. In summary, samples were analyzed with a Waters ACQUITY ultra- UPLC and a Thermo Scientific Q-Exactive high resolution/accurate mass spectrometer, interfaced with a heated electrospray ionization (HESI-II) source and Orbitrap mass analyzer operated at 35,000 mass resolution. The sample extracts were dried then reconstituted in solvents compatible to each of the four methods applied: acidic positive ion conditions, chromatographically optimized for more hydrophilic compounds; acidic positive ion conditions, but chromatographically optimized for more hydrophobic compounds; basic negative ion optimized conditions, using a separate dedicated C18 column; negative ionization following elution from an HILIC column. Compounds were identified by comparison to library entries of purified standards or recurrent unknown entities, based on retention time, parent ion accurate mass, and MS/MS fragmentation spectrum to an authentic standard, [28] which represents Tier 1 identification [29].

### 2.6. Statistical Analysis

Descriptive statistics were used to characterize the included study population. To analyze the association between urine metabolite levels and AMD case-control status, we computed multivariable logistic regression models with a binary outcome (AMD, including subjects with any stage of AMD (1) and control (0)). Each model included a single metabolite as a continuous variable, adjusting for age, gender, BMI, and use of AREDS vitamins supplementation at the time of study inclusion. Initially, models were computed for the two study cohorts (Boston, US and Coimbra, Portugal) separately, and then these were meta-analyzed using a fixed-effects method [30]. For each model, in addition to the meta-analysis *p*-values, we provide the odds ratio of each metabolite for both cohorts. In summary, the odds ratio represents the effect size of one-unit increase (i.e., standard deviation) of the urine metabolite levels on the odds of AMD case status (versus control).

To further assess the association between urine metabolite levels and AMD, severity stage of disease was also considered as an outcome. For this analysis, we used multivariate ordinal logistic regression models, with an ordinal outcome: control (0), early (1), intermediate (2), and late (3) stage AMD. Again, the analyses for each metabolite (adjusted for age, gender, BMI, and AREDS supplementation) were performed for each cohort and meta-analyzed using a fixed-effects model [30]. In these models, the odds ratio provided for each cohort represents the effect size of a one-unit increase (i.e., standard deviation) of the urine metabolite levels on the odds of having a more severe stage of disease.

All analyses were conducted in R, version 4.1.1. Each individual was included only once and if the two eyes of an individual had a different AMD stage, the worst eye was considered to define the stage of the individual. *p*-values < 0.01 are reported.

## 3. Results

A total of 484 subjects were included; 185 individuals from Boston, US and 299 from Coimbra, Portugal—Table 1. After quality control (see the Methods section) and exclusion of exogenous metabolites, information on 710 endogenous urinary metabolites was considered.

### 3.1. Urinary Metabolites Associated with AMD

First, we performed logistic regression analyses for the Boston and Coimbra cohorts (Appendix A) to assess associations between urinary metabolomic levels and AMD. Then, results were combined by meta-analysis. Based on a *p*-value < 0.01, no urinary metabolites were found to be significantly associated with AMD case-control status.

### 3.2. Urinary Metabolites Associated with AMD Severity Stages

As described above, to assess urinary metabolomic profiles of AMD from our two cohorts, we conducted ordinal logistic regression analyses first for the Boston and Coimbra cohorts separately (Appendix A), and then the results were combined by meta-analysis. Meta-analysis identified six metabolites differing significantly across severity stages (Table 2), two of which (sphingosine and phosphoethanolamine) were previously reported by our group to also differ in the plasma of patients with AMD and controls and across stages of disease [18].

## 4. Discussion

We present a cross-sectional study evaluating associations between urinary metabolomic profiles assessed by mass spectrometry and AMD. Our results did not show any statistically significant differences when comparing AMD patients as a group to controls. However, when looking at the disease as a spectrum, weak statistical differences were seen. Of note, two of the urinary metabolites identified (sphingosine and phosphoethanolamine) have been previously shown by our group to also differ in the plasma of patients of AMD compared to controls and across severity stages [18].

Urine is easy to obtain and a non-invasive biological sample, which has been used in multiple fields to derive metabolomic biomarkers of multifactorial diseases. [19,20,21,22] In this study, however, urine MS metabolomics was unable to separate patients with AMD from controls. The cohort included here has been previously well-characterized by our group and shown strong differences in the plasma metabolomic profiles of AMD patients and controls [18]. Thus, our results suggest that this biofluid is not as well-suited as plasma to the study of AMD. This has been previously suggested by our work using a different metabolomics’ technique, nuclear magnetic resonance spectroscopy, NMR [15], where less significant results were also seen than those obtained for NMR with plasma [14]. Our current results, however, are difficult to compare with our urine NMR work, as NMR identified some of the lipid signals differing across stages of disease, but these were not possible to link with specific named metabolites [15].

Despite this, when considering a nominal *p*-value (*p* < 0.01), we identified differences in the urinary metabolomic profiles across AMD stages, with two of the identified metabolites in common with those described by our group as differing in the plasma of patients with AMD compared to controls and across stages of disease [18]. In particular, differences in urinary levels of sphingosine, a sphingolipid, were observed. This is also in agreement with a recent study on differentially expressed genes in AMD combining microarray information from RPE-choroid, retinal tissue, and blood samples, which identified an enrichment in the sphingolipid pathway [31]. Sphingolipids are a structurally diverse class of lipids that play an important role in cell membranes, but also participate in signal transduction and cell recognition, representing versatile signaling molecules that regulate multiple physiological and pathological processes [32]. In particular, sphingosine participates in apoptosis and induces cell death in response to multiple inducers, such as oxidative stress, [32,33] a known central pathophysiologic mechanism in AMD [34,35]. Indeed, in the retina, studies have reported that sphingosine promotes the death of photoreceptors and amacrine cells [32,33] and to be involved in AMD progression [32]. Despite this, data on sphingolipid receptors and metabolic enzymes in the retina are still scarce [32].

The observed differences in urinary phosphoethanolamine across stages of disease were also present in plasma [18], and our recent study on metabolic quantitative trait loci (mQTL) in AMD [36] also revealed that the most significant mQTL were seen in polymorphisms in the *LIPC* gene with levels of phosphatidylethanolamines. Phosphoethanolamines are glycerophospholipids [37] and appear to have a dominant role in the vertebrate retina. In particular, they are thought to be involved in the transport of visual pigments. All our work to date on MS metabolomics of AMD has consistently shown differences in the glycerophospholipid pathway [18], and a recent study in human donor eyes also described significant differences in PE metabolites in eyes with AMD compared to controls [38].

This study has important limitations that should be noted. First, this is a cross-sectional study, and even though we have described changes across stages of AMD, longitudinal studies are better suited to address if urine metabolites are associated with progression over time. Even though our sample size is relatively large for metabolomics’ studies, and we have described in this same cohort very significant changes in plasma metabolomic profiles [18], urinary signals may be weaker. Thus, our lack of significant findings comparing AMD patients with controls may represent a true lack of association or, less likely, low power to detect weak differences. Additionally, most of our significant findings in plasma were lipid metabolites, and these are often difficult to identify in urine. Importantly, in this work, we report nominal significant p-values, but we did not adjust for multiple comparisons, which increases the likelihood of having false positive associations. This is one of the reasons why we focused our discussion on urinary metabolites that are in agreement with our plasma work [18]. Due to the small number of patients with late AMD, we also did not perform analysis comparing urinary profiles of exudative and dry AMD. This would be interesting as it is likely that these differ, and other authors have described different levels of phosphoethanolamines in patients with neovascular AMD and controls [30]. The external validity of our study might also be limited because our cohorts were nearly all Caucasian subjects. This is related in part to the epidemiology of AMD [39], and in part to the population served by both enrolling sites, two tertiary care hospitals. Despite these limitations, to our knowledge, this is the first study assessing urinary metabolomic changes in AMD measured by MS, and we followed a pre-established, prospectively designed protocol consistent in our two study sites. Our samples were collected after fasting and immediately stored for metabolomic profiling, which identifies metabolites using a chemocentric approach with standards for each identified metabolite and was performed using a state-of-the-art platform that covers a wide range of the metabolome.

## 5. Conclusions

In conclusion, this study suggests that urinary metabolomic profiles have weak statistical differences across stages of AMD, with two of the metabolites identified in common with those previously described by our group as significant in plasma—sphingosine and phosphoethanolamine. These findings suggest that even though urine metabolomics associations with AMD are not strong, these may represent future biomarkers of AMD, which with further research may have the potential to be applicable to clinical practice.

## Figures and Tables

**Table 1 jcm-11-00940-t001:** Clinical and demographic characteristics of the included population.

Boston, US					
	Control	Early AMD	Intermediate AMD	Late AMD	Total
Number, *n* (%)	45 (24.3)	32 (17.3)	62 (33.5)	46 (24.9)	185 (100.0)
Age, Mean ± SD	72.1 ± 8.5	73.7 ± 6.9	77.5 ± 6.7	81.3 ± 7.8	76.6 ± 8.1
Female Gender, n (%)	27 (60.0)	21 (65.6)	45 (72.6)	20 (54.1)	119 (64.3)
BMI, Mean ± SD	27.1 ± 4.4	26.5 ± 4.2	27.6 ± 5.5	26.9 ± 4.5	27.1 ± 4.8
Race/Ethnicity, *n* (%) - White - Black - Hispanic - Asian	42 (93.3) 1 (2.2) 2 (4.4) 0 (0.0)	30 (93.8) 0 (0.0) 2 (6.3) 0 (0.0)	60 (96.8) 0 (0.0) 0 (0.0) 2 (3.2)	43 (93.5) 0 (0.0) 3 (6.5) 0 (0.0)	175 (94.6) 1 (0.5) 7 (3.8) 2 (1.1)
Smoking, *n* (%) - Non-smoker - Ex-smoker - Smoker	24 (53.3) 19 (42.2) 2 (4.4)	18 (56.3) 14 (43.8) 0 (0.0)	26 (41.9) 33 (53.2) 3 (4.8)	16 (34.8) 30 (65.2) 0 (0.0)	84 (45.4) 96 (51.9) 5 (2.7)
On AREDS Supplementation (Yes), *n* (%)	2 (4.4)	2 (6.3)	45 (72.6)	31 (67.4)	80 (43.2)
**Coimbra, Portugal**					
Number, *n* (%)	50 (16.7)	57 (19.1)	139 (46.5)	53 (17.7)	299 (100.0)
Age, Mean ± SD	72.5 ± 5.1	75.0 ± 6.1	80.4 ± 7.5	85.7 ± 6.9	79.0 ± 8.0
Female Gender, n (%)	32 (64.0)	34 (59.6)	96 (69.1)	31 (58.5)	193 (64.5)
BMI, Mean ± SD	27.0 ± 4.6	27.2 ± 4.3	27.6 ± 4.6	26.5 ± 4.3	27.2 ± 4.5
Race/Ethnicity, *n* (%) - White - Black - Hispanic - Asian	50 (100.0) 0 (0.0) 0 (0.0) 0 (0.0)	57 (100.0) 0 (0.0) 0 (0.0) 0 (0.0)	137 (98.6) 2 (1.4) 0 (0.0) 0 (0.0)	52 (98.1) 1 (1.9) 0 (0.0) 0 (0.0)	296 (99.0) 1 (1.0) 0 (0.0) 0 (0.0)
Smoking, *n* (%) - Non-smoker - Ex-smoker - Smoker	40 (80.0) 10 (20.0) 0 (0.0)	49 (86.0) 8 (14.0) 0 (0.0)	123 (88.5) 16 (11.5) 0 (0.0)	38 (71.7) 14 (26.4) 1 (1.9)	250 (83.6) 48 (16.1) 1 (0.3)
On AREDS Supplementation (Yes), *n* (%)	0 (0.0)	1 (1.8)	2 (1.4)	8 (15.1)	11 (3.7)

Legend: *n*—number, SD—standard deviation, BMI—body mass index, AMD—age-related macular degeneration, AREDS—Age-Related Eye Disease Study.

**Table 2 jcm-11-00940-t002:** Urinary metabolites differing significantly across severity stages (*p* < 0.01).

Metabolite	HMDB	Super Pathway	Sub Pathway	Odds Ratio Boston	Odds Ratio Portugal	*p*-Value Meta-Analysis	Significant in Plasma [18]
Indoleacetylglutamine	HMDB0013240	Amino Acid	Tryptophan Metabolism	0.918	0.349	0.0022	No
11-ketoetiocholanolone sulfate	NA	Lipid	Androgenic Steroids	2.021	1.753	0.0040	Not Measured in Plasma
Tetrahydrocortisol sulfate (2)	NA	Lipid	Corticosteroids	4.853	1.280	0.0051	Not Measured in Plasma
Adipate (C6-DC)	HMDB0000448	Lipid	Fatty Acid, Dicarboxylate	0.566	0.517	0.0061	Not Measured in Plasma
Sphingosine	HMDB0000252	Lipid	Sphingosines	0.437	0.664	0.0063	**Yes**
Phosphoethanolamine	HMDB0000224	Lipid	Phospholipid Metabolism	1.767	1.752	0.0100	**Yes**

Legend: HMDB—Human Metabolome Database identifier.

## Data Availability

All data can be made available upon request.

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
