# Peer review of "Urinary Mass Spectrometry Profiles in Age-Related Macular Degeneration"

_jcm, 2022, doi:10.3390/jcm11040940_

Round 1

Reviewer 1 Report

In this study, the authors compare urine metabolomic profile analyzed by mass spectrometry between patients with AMD and controls, and across severity stages of AMD.

In my opinion, this is an outstanding work performed by a very well consolidated group which is the leader in the field of metabolomics in AMD.

This study delivers very interesting findings, it is written in a very concise and understandable manner, data are presented in a meticulous and intelligible fashion, and results support the conclusions displayed. Also, the limitations of the study are thoroughly disclosed.

Overall, I would like to congratulate the authors for their magnificent work. 

Here are my specific comments:

  • Line 17. The text states “Urine is a non-invasive biofluid and (…)”. I encourage the authors to rephrase this sentence.
  • Line 135. Authors state that “To analyze the association between urine metabolite levels and AMD case-control status, we computed multivariable logistic regression models with a binary outcome (AMD (1) and control (0))”. Please specify in this section which patients where included in the AMD group (what stage was considered AMD in this analysis).
  • Line 230. Please briefly elaborate the relationship between phosphoethanolamine and AMD as it has been done with sphingosine.

Author Response

Reviewer 1

In this study, the authors compare urine metabolomic profile analyzed by mass spectrometry between patients with AMD and controls, and across severity stages of AMD. In my opinion, this is an outstanding work performed by a very well consolidated group which is the leader in the field of metabolomics in AMD. This study delivers very interesting findings, it is written in a very concise and understandable manner, data are presented in a meticulous and intelligible fashion, and results support the conclusions displayed. Also, the limitations of the study are thoroughly disclosed. Overall, I would like to congratulate the authors for their magnificent work. Here are my specific comments:

  • Line 17. The text states “Urine is a non-invasive biofluid and (…)”. I encourage the authors to rephrase this sentence.

We appreciate this suggestion and have modified this sentence to “Urine is a biofluid that can be obtained non-invasively and, in other fields, urine metabolomics has been proposed as a feasible alternative to plasma biomarkers.”

  • Line 135. Authors state that “To analyze the association between urine metabolite levels and AMD case-control status, we computed multivariable logistic regression models with a binary outcome (AMD (1) and control (0))”. Please specify in this section which patients were included in the AMD group (what stage was considered AMD in this analysis).

In this analysis we compared all patients with AMD (i.e. with any stage of AMD) with our control group. This information was added to the revised version of our manuscript:

“To analyze the association between urine metabolite levels and AMD case-control status, we computed multivariable logistic regression models with a binary outcome (AMD, including subjects with any stage of AMD (1) and control (0)).”

  • Line 230. Please briefly elaborate the relationship between phosphoethanolamine and AMD as it has been done with sphingosine.

We appreciate this point raised by the Reviewer. The reason we did not elaborate more on phosphoethanoloamine and associations with AMD is because this has been done in our previous work. Considering this suggestion, however, we added the following to our revised manuscript:

“The observed differences in urinary phosphoethanolamine across stages of disease were also present in plasma [18] and our recent study on metabolic quantitative trait loci (mQTL) in AMD [34] also revealed that the most significant mQTL were seen in polymorphisms in the LIPC gene with levels of phosphatidylethanolamines. Phosphoethanolamines are glycerophospholipids,[35] and appear to have a dominant role in the vertebrate retina. In particular, they are thought to be involved in the transport of visual pigments. All our work to date on MS metabolomics of AMD has consistently shown differences in the glycerophosphospholipid pathway [18] and a recent study in human donor eyes also described significant differences in PE metabolites in eyes with AMD compared to controls.[36]”

Reviewer 2 Report

Lains and colleagues describe urine mass spectrometry profiles of patients with AMD. This is a great first start for the use of urine biomarkers that can be associated with AMD. I agree that urine is a novel approach to assess metabolites that may predict AMD, my concern over being able to rely on urine is as you have noted that only a few metabolites were found with significant disease. Perhaps other forms of elimination (ie the liver) account for the limited biomarkers. Blood samples may offer a more sensitive screen if you will, regardless, this is interesting work. Lastly, without a Bonferroni correction in the setting of analyzing 700 variables may falsely identify significant metabolites.

Introduction:

The first two paragraphs, although, nicely written, could be condensed and then would recommend discussing more on the differences in urine and blood metabolomics and why urine would be favorable (other than a less invasive target).

Methods:

It would have been interesting to report the analyses separately given the significant difference in geography between the two cohorts.

Additionally, with over 700 metabolites examined, a Bonferroni correction may have been needed as 0.01 is not enough.

Discussion: More mention of the metabolites not measured in plasma would be interesting. Perhaps these are markers urine may be uniquely able to identify and offer an advantage to blood samples.

Author Response

Reviewer 2

Lains and colleagues describe urine mass spectrometry profiles of patients with AMD. This is a great first start for the use of urine biomarkers that can be associated with AMD. I agree that urine is a novel approach to assess metabolites that may predict AMD, my concern over being able to rely on urine is as you have noted that only a few metabolites were found with significant disease. Perhaps other forms of elimination (ie the liver) account for the limited biomarkers. Blood samples may offer a more sensitive screen if you will, regardless, this is interesting work. Lastly, without a Bonferroni correction in the setting of analyzing 700 variables may falsely identify significant metabolites.

We thank the Reviewer for this comment. We agree that urine does not appear to be the most suited biomarker for AMD and that plasma seems to provide more robust findings. This is highlighted in our manuscript:

  • Abstract: “While there are fewer urinary metabolites associated with AMD than plasma metabolites, this study identified some differences across stages of disease that support previous work performed with plasma, thus highlighting the potential of these metabolites as future biomarkers for this blinding disease.”
  • Conclusion: “In conclusion, this study suggests that urinary metabolomic profiles have weak statistical differences across stages of AMD, with two of the metabolites identified in common with those previously described by our group as significant in plasma - sphingosine and phosphoethanolamine. These findings suggest that even though urine metabolomics associations with AMD are not strong (…).”

We also agree that the fact that by not accounting for multiple testing these results may represent false associations. This is the reason why we cautiously describe that we are presenting nominal p-values and why we focused our discussion and results around those that are in agreement with what we have observed in plasma. Indeed, in our plasma work (namely reference 18) we accounted for multiple comparisons. Considering this suggestion, we have further expanded our limitations’ section: “Importantly, in this work, we report nominal significant p-values, but we did not adjust for multiple comparisons, which increases the likelihood of having false positive associations. This is one of the reasons why we focused our discussion on urinary metabolites that are in agreement with our plasma work.[18]”

Introduction: The first two paragraphs, although, nicely written, could be condensed and then would recommend discussing more on the differences in urine and blood metabolomics and why urine would be favorable (other than a less invasive target).

We appreciate this comment and have added the following to the introduction of our revised manuscript: “Even though most research on metabolomics to derive biofluid disease biomarkers has been performed in blood samples,[19–22] urine has been increasingly used as a biomatrix for metabolomic profiling in other multifactorial diseases.[23] Urine is a very appealing biofluid because it can be collected non-invasively, is produced regularly and in abundant quantities, and gives a time-averaged representation of an individual’s recent homeostatic status.[24]”

Methods: It would have been interesting to report the analyses separately given the significant difference in geography between the two cohorts.

We appreciate this comment by the Reviewer. All the separate analyses for Coimbra, Portugal and Boston, US have been submitted as supplementary tables.

Additionally, with over 700 metabolites examined, a Bonferroni correction may have been needed as 0.01 is not enough.

As mentioned, we agree that by not accounting for multiple testing our results may represent false associations. This is the reason why we cautiously describe that we are presenting nominal p-values and why we focused our discussion and results around those that are in agreement with what we have observed in plasma. Indeed, in our plasma work (namely reference 18) we accounted for multiple comparisons. Considering this suggestion, we have further expanded our limitations’ section: “Importantly, in this work, we report nominal significant p-values, but we did not adjust for multiple comparisons, which increases the likelihood of having false positive associations. This is one of the reasons why we focused our discussion on urinary metabolites that are in agreement with our plasma work.[18]”

Discussion: More mention of the metabolites not measured in plasma would be interesting. Perhaps these are markers urine may be uniquely able to identify and offer an advantage to blood samples.

We appreciate this point raised by the Reviewer and we agree that metabolites identified in urine could be of unique value. However, considering that in our study we did not account for multiple comparisons, as mentioned above we believe that we cannot confidently make a discussion about the metabolites that were solely identified in urine. Of note, we are currently working on a manuscript to specifically look at similarities and differences in metabolites identified in urine and plasma (regardless of disease associations), and hope to report these results soon.

Reviewer 3 Report

Is an interesting topic, a new topic, an original idea.

the material and method should be explained more detail about urinary matalobolomic detailes !True, urine is easy to get, but determinations are costly!

The results  are as expected!

Author Response

Reviewer 3

Is an interesting topic, a new topic, an original idea. The material and method should be explained more detail about urinary metabolomic details. True, urine is easy to get, but determinations are costly! The results are as expected!

We agree that more details should be provided about urinary metabolomics, and have added the following to the revised version of our manuscript: “Non-targeted MS analysis was performed by Metabolon, using Ultrahigh Performance Liquid Chromatography-Tandem MS (UPLC-MS/MS), according to protocols that have been previously described [29]. In summary, samples were analyzed with a Waters ACQUITY ultra- UPLC and a Thermo Scientific Q-Exactive high resolution/accurate mass spectrometer, interfaced with a heated electrospray ionization (HESI-II) source and Orbitrap mass analyzer operated at 35,000 mass resolution. The sample extracts were dried then reconstituted in solvents compatible to each of the four methods applied: acidic positive ion conditions, chromatographically optimized for more hydrophilic compounds; acidic positive ion conditions, but chromatographically optimized for more hydrophobic compounds; basic negative ion optimized conditions, using a separate dedicated C18 column; and negative ionization following elution from a HILIC column. Compounds were identified by comparison to library entries of purified standards or recurrent unknown entities, based on retention time, parent ion accurate mass and MS/MS fragmentation spectrum to an authentic standard,[29] which represents Tier 1 identification.[30]”

We would agree that unfortunately metabolomic profiling remains an expensive technique. However, the cost the same for urine and plasma. We expect that in the future with the more widespread use of metabolomics the costs will become more affordable, but that is certainly a current limitation of this approach.